# Successful Rescue of Synthetic AcMNPV with a ~17 kb Deletion in the C1 Region of the Genome

**DOI:** 10.3390/v14122780

**Published:** 2022-12-13

**Authors:** Yijia Guo, Hengrui Hu, Han Xiao, Fei Deng, Jiang Li, Manli Wang, Zhihong Hu

**Affiliations:** 1Centre for Biosafety Mega-Science, Wuhan Institute of Virology, Chinese Academy of Sciences, Wuhan 430071, China; 2University of the Chinese Academy of Sciences, Beijing 100049, China; 3State Key Laboratory of Virology and National Virus Resource Centre, Wuhan Institute of Virology, Chinese Academy of Sciences, Wuhan 430071, China

**Keywords:** baculoviruses, synthetic genome, non-essential genes, genome reduction

## Abstract

Baculoviruses have been widely used as expression vectors. However, numerous genes in the baculoviral genome are non-essential for cellular infection and protein expression, making the optimisation of baculovirus expression vectors possible. We used a synthetic biological method to reduce the number of genes in a partial region of the autograph californica multiple nucleopolyhedrovirus (AcMNPV), the most widely used baculovirus expression vector. The C1 region of the AcMNPV is 46.4 kb and is subdivided into B1, B2, and B3 fragments. We first designed modified B1, B2, and B3 fragments by deleting the non-essential genes, and then synthesised complete viral genomes containing either individual modified B fragments or joint modified B fragments through transformation-related recombination in yeast. The synthetic genomes were then transfected into Sf9 cells to rescue the progeny viruses and test their infectivity. The design-build-test cycle was repeated until the ultimately rescued virus could produce progeny viruses efficiently. Finally, AcMNPV-Syn-mC1-1.1 by deleting approximately 17.2 kb, including 20 ORFs, in the C1 region, was obtained. This is essential to the synthesis of a minimal AcMNPV genome that can generate infectious progeny viruses and can be further used to optimise the foundation of baculovirus expression vectors.

## 1. Introduction

Baculoviruses are insect-specific pathogens with a circular double-stranded DNA (dsDNA) genome and have been widely used as biological control agents against fundamental agricultural and forestry pests [1,2]. In addition, they are potent expression vectors owing to their high expression, high abundance, and relatively complete post-translational modification of proteins. Since the first report of human IFN-β expression using the baculovirus expression vector system (BEVS) in 1983 [3], many proteins have been expressed via this system. Moreover, certain biological products have been approved by the Food and Drug Administration and marketed. For example, FluBlok^®^ and Cervarix^®^ vaccines were produced using baculovirus expression systems to guard against influenza and human papilloma virus, respectively, and have been commercialised [4]. In addition, an AAV vector was produced from Glybera, the first gene therapy product approved in Europe, using BEVS [5].

Insects are the natural hosts of baculoviruses. Because insects appear periodically, baculoviruses have developed a unique life cycle with two different phenotypes of progeny viruses: occlusion-derived viruses (ODVs) and budded viruses (BVs). ODVs are occluded in occlusion bodies (OBs) which serve as shelters until the OBs are digested by insect larvae, and the ODVs are released from OBs to initiate primary infection. BVs are produced from infected midgut epithelial cells and spread the infection to other insect tissues. Therefore, ODVs are responsible for initiating and spreading infections in insect populations, whereas BVs are responsible for systemic infection within infected larvae [6]. Although these two phenotypes are necessary for the natural life cycle of baculoviral infections, the BEV mainly relies on the productivity of BV.

The genomes of more than a hundred different baculoviruses have been sequenced; they range from 80 to 180 kb and encode 90 to 180 genes [7]. Numerous genes in the baculovirus genome are non-essential for BV production in vitro. For instance, baculovirus encodes 10 *per os* infectivity factors (PIFs), which are essential for oral infection in larvae. However, the deletion of most PIFs has no impact on BV production in cell culture [8]. In fact, the deletion of certain genes, such as *chitinase* and *cathepsin*, enhances the properties of the baculoviral expression system [9,10]. Therefore, the deletion of non-essential genes from the baculovirus genome to make it an optimized vector with a larger capacity is desirable. 

Autograph californica multiple nucleopolyhedrovirus (AcMNPV) is the most well-studied baculovirus and has been widely used to generate baculoviral vectors. It contains a genome of approximately 134 kb and encodes 156 genes [11]. We previously synthesised an AcMNPV genome (AcMNPV-WIV-Syn1) based on the three-step transformation-associated recombination (TAR) in yeast [12]. TAR based on co-transformation of overlapping gene fragments and vector DNAs into *Sacharomyces cerevisae*, where these fragments and the vector DNAs are joined by homologous recombination. This method was used to synthesis a *Mycoplasma genitalium* genome [13]. During synthesis, the genome was first amplified in 45 A-level fragments (A1–A45) of ~3 kb, then assembled sequentially into 9 B-level fragments (B1–B9) of ~15 kb, three C-level fragments (C1–C3) of ~45 kb, and finally a ~145 kb full genome. The rescued synthetic virus showed biological properties similar to those of the wild-type parental virus [12]. Recently, we systemically studied the impact of deleting individual AcMNPV genes on BV production and proposed a schematic design for generating a minimal AcMNPV genome [14]. These developments allow for the construction of a minimised AcMNPV genome that deletes all unnecessary genes for BV production and retains suitable reproduction in cell culture. 

In this study, as the first step to generate a minimised AcMNPV genome, we focused on a 46.4 kb C1 region of the AcMNPV genome and used synthetic biology to reduce the region. The classical design-build-test synthetic approach was adopted. We first designed modified B1–B3 fragments by deleting all non-essential ORFs and synthesised the first version of modified B fragments by TAR in yeast. The modification effect was then tested by constructing an AcMNPV bacmid and rescuing the virus. After a few rounds of the design-build-test cycle, a synthetic AcMNPV-Syn-mC1-1.1 genome with a ~17 kb deletion that had the ability to produce infectious BVs after rescuing the transfected insect cells was constructed.

## 2. Materials and Methods

### 2.1. Cells and Viruses

The Sf9 insect cell line, originally obtained from the fall armyworm *Spodoptera frugiperda*, was cultured at 27 °C in Grace’s medium (Invitrogen America) supplemented with 10% foetal bovine serum (Gibco, Carlsbad, CA, USA) [15]. AcMNPV-WIV-Syn1 [12] was used as the parental virus in this study. *Saccharomyces cerevisiae* strain VL6−48N *(MAT alpha, his3-Δ200, trp1-Δ1, ura3-Δ1, lys2, ade2−101, met14*), *Escherichia coli* strain EPI300 carrying an inducible *trfA* gene, and the TAR cloning vector pGF were obtained from Prof. Gengfu Xiao of the Wuhan Institute of Virology, China Academy of Science [16]. Vector pGF-*egfp* was constructed previously and maintained in the laboratory [12].

### 2.2. Construction and Synthesis of Modified Genomes

The C1 region in AcMNPV-WIV-Syn1 is 46.4 kb containing 55 ORFs. Based on the individual gene knockout results [14] and transcriptome analysis data [17], the initial design of deleting all non-essential genes was made by constructing the first version of modified B1–B3. For example, the modified sub-fragments of B1 were first constructed by overlap PCR using the primers listed in Table 1 and the previously constructed B1 [12] as the template. Then, modified B1 (mB1-1.0) was synthesised through TAR in yeast using the two PCR fragments and the pGF vector. The full genome Syn-mB1-1.0 was generated by co-transfection of mB1-1.0, B2, B3, C2, C3, and pGF-*egfp* vector DNA into *S. cerevisiae* VL6−48N and was synthesised via TAR. Similar methods were used to generate Syn-mB2-1.0, and Syn-mB3-1.0. In the meantime, three modified B-level fragments (mB1-1.0, mB2-1.0, and mB3-1.0), C2, C3, and pGF-*egfp* vector were used to synthesise Syn-mC1-1.0. Synthetic genomes were extracted from yeast cells and then electroporated into the EPI300 *E. coli* strain. After confirmation by PCR and restriction enzyme analysis, the correct genomes were further analysed. After evaluation using transfection and infection assays, the necessary redesigns were made, and new versions of the synthesised genome were generated via the aforementioned methods.

### 2.3. Transfection and Infection Assays

Bacmid DNA was extracted from *E. coli* using methods developed for large plasmids (Instruction Manual of Bac-to-Bac System/Life Technologies, Carlsbad, CA, USA). Sf9 cells (2 × 10^6^) were transfected with 5 μg of each bacmid DNA using Cellfectin (Invitrogen Carlsbad, CA, USA). The cells were observed under a fluorescence microscope (Invitrogen, Carlsbad, CA, USA) at different time points post-transfection (p.t.). Supernatants of transfected cells were collected at 120 h p.t. and used to infect fresh Sf9 cells. At 120 h post-infection (p.i.), the supernatants were collected, and the titre of BV was determined via the fluorescence of the infected cells using an end-point dilution assay (EPDA). To characterise the BV productivity of the rescued viruses, Sf9 cells (1 × 10^6^) were infected with the rescued viruses at a multiplicity of infection (MOI) of 1, supernatants were collected at 96 h p.i., and the BV titres were measured using EPDA.

### 2.4. Restriction Enzyme Digestion Analysis and Genome Sequencing of AcMNPV-Syn-mC1-1.1

The DNA of the final synthetic bacmid AcMNPV-Syn-mC1-1.1 and AcMNPV-WIV-Syn1 was extracted from *E. coli* using the Plasmid Mini Kit (QIAGEN, Hilden, Germany). Three restriction enzymes: BglII, XhoI, and SacII (Takara, Osaka, Japan), were used to perform restriction enzyme digestion assays of bacmid AcMNPV-Syn-mC1-1.1 and AcMNPV-WIV-Syn1. The simulation of the enzyme digestion results was predicted using the computer program Snapgene (Insightful Science, San Diego, USA, version 3.2.1) [12]. The DNA of AcMNPV-Syn-mC1-1.1 was subjected to the standard process of the BGISEQ/MGISEQ NGS system (BGI) for genome sequencing. The data were verified using FastQC (version 0.11.9), and the reads were aligned to the reference sequence of AcMNPV-WIV-Syn1 (NCBI accession No. KY792989.1) using Bowtie2 (http://sourceforge.net/projects/bowtie-bio/files/ (accessed on 12 November 2022), version 2.3.4.3), allowing for a single mismatch [18]. The genome sequence of AcMNPV-Syn-mC-1.1 was deposited in GenBank under the accession no. OP627104.

### 2.5. One-Step Growth Curve

2 × 10^6^ Sf9 cells were infected with AcMNPV-Syn-mC1-1.1 or AcMNPV-WIV-Syn1 at an MOI of 1. Supernatants were collected at 0, 24, 48, and 72 h p.i., and the titres of progeny BVs were determined using EPDA. Each experiment was performed in triplicates. Student’s *t*-test was used for significant difference analysis between the titres of the two viruses at each time point using the computer program Graphpad Prism v8 (Insightful Science, San Diego, USA,) [12].

### 2.6. Electron Microscopy

Sf9 cells (1 × 10^6^) were infected with AcMNPV-WIV-Syn1 and AcMNPV-Syn-mC1-1.1 at an MOI of 5. At 24, 48, and 72 h p.i., the cells were fixed with 2.5% glutaraldehyde for more than 2 h, and ultrathin sections were prepared as previously described [19]. Images were obtained using an electron microscope Tecnai G^2^ 20 TWIN (FEI, Carlsbad, CA, USA) at an accelerating voltage of 200 kV.

## 3. Results

### 3.1. Synthesis and Characterization of the Version 1.0 of Modified Genomes 

Previously, AcMNPV-WIV-Syn1 was synthesised based on three C fragments (C1–C3) to form the entire AcMNPV genome [12]. Each C fragment was generated from three B fragments (Figure 1A)**.** To generate a functional genome with minimalized C1 region, a design-synthesis-test cycling approach was used in our study (Figure 1B). According to previous studies [14,17], we designed the first version of modified B1–B3 (mB1-1.0, mB2-1.0, and mB3-1.0) by deleting non-essential genes but retaining other genes, including the required transcriptional signals. The fragments were generated by overlap extension PCR and TAR in yeast, and the correct fragments were used individually to synthesise the genomes of AcMNPV-Syn-mB1-1.0, AcMNPV-Syn-mB2-1.0, and AcMNPV-Syn-mB3-1.0. In addition, the three modified fragments: mB1-1.0, mB2-1.0, and mB3-1.0, were combined to synthesise the AcMNPV-Syn-mC1-1.0 genome, where the entire C1 sequence was modified. All the synthesised genomes were transfected to Sf9 cells, and the supernatants were collected for infection to test their capacity for BV production, which was then compared with that of the parental virus AcMNPV-WIV-Syn1. When the synthesised virus showed infectivity similar to that of AcMNPV-WIV-Syn-1, this version of the design was retained. Otherwise, the fragments were redesigned, synthesised, and tested (Figure 1).

To be specific, the original B1 fragment is approximately 14.7 kb and contains 17 ORFs, among which six genes (*ptpase, lef-2, p78, pk-1, ac11, lef-1*) are essential for budded virus production [14]. Therefore, modified mB1-1.0 was designed to retain these six genes and the homologous repeat region 1a (*hr1a*), and approximately 7.7 kb sequences were deleted. The original B2 contains 18 ORFs including four essential genes (*ac19, pkip, dbp, ac34*), and these four genes and *ac18*, *pnk* were reserved to generate mB2-1.0, which deleted approximately 11.2 kb. Similarly, mB3-1.0 was designed which deleted 5.4 kb including six ORFs (*lef-12, gta, ac43, ac44, ac45, odv-e66*) (Figure 2A). 

After synthesising AcMNPV-Syn-mB1-1.0, AcMNPV-Syn-mB2-1.0, AcMNPV-Syn-mB3-1.0, and AcMNPV-Syn-mC1-1.0 genomes, transfection and infection assays were conducted in Sf9 cells, and the results showed that only AcMNPV-Syn-mB3-1.0 efficiently produced infectious BVs, as evidenced by fluorescently infected cells (Figure 2B). Infection with AcMNPV-Syn-mB1-1.0 yielded clumps of fluorescent cells, suggesting the production of low levels of infectious BVs. No fluorescent cells were observed at 96 h p.i. in the AcMNPV-Syn-mB2-1.0 sample, indicating that infectious BV was not produced. AcMNPV-Syn-mC1-1.0 also showed no infective BV production. The above tests showed that only AcMNPV-Syn-mB3-1.0 was functionally successful, whereas the others needed to be redesigned. 

### 3.2. Redesign and Synthesis of AcMNPV-Syn-mB1-1.1, AcMNPV-Syn-mB2-1.1, and AcMNPV-Syn-mC1-1.1

Next, we redesigned the modifications to B1 and B2. Fragment mB1-1.1 was designed by repairing four ORFs (*ac12, ac13, egt, odv-e26*) to mB1-1.0; and mB2-1.1 by repairing four ORFs (*ac17, ac26, iap-1, lef-6*) to mB2-1.0 (Figure 3A). Fragments mB1-1.1 and mB2-1.1 were synthesised using the same methods as shown in Figure 1B, and the synthesised AcMNPV-Syn-mB1-1.1 and AcMNPV-Syn-mB2-1.1 genomes were generated using mB1-1.1 and mB2-1.1, respectively. Transfection and infection assays showed that the two genomes could generate infectious progeny viruses (Figure 3B). Then, AcMNPV-Syn-mC1-1.1 was synthesised with mB1-1.1, mB2-1.1, mB3-1.0, and the fragments of the rest of the genome. The results showed that this genome produce infectious progeny viruses. Parental AcMNPV-WIV-Syn1 was used as a positive control in the transfection and infection assays (Figure 3B). 

To characterise the BV productivity of the rescued viruses, Sf9 cells were infected with AcMNPV-Syn-mB1-1.1, AcMNPV-Syn-mB2-1.1, AcMNPV-Syn-mB3-1.0, AcMNPV-Syn-mC1-1.1, and AcMNPV-WIV-Syn1 at an MOI of 1, the supernatant was collected at 96 h p.i., and BV titres were measured. As shown in Table 2, the BV titre of the positive control AcMNPV-WIV1-Syn1 reached 6.22 × 10^7^ TCID_50_/mL. The titres of AcMNPV-Syn-mB1-1.1 and AcMNPV-Syn-mB3-1.0 were not significantly different from those of the parental virus; however, the titre of AcMNPV-Syn-mB2-1.1 (4.58 × 10^6^ TCID_50_/mL) was significantly lower than that of the parental virus (*p* < 0.01). The BV titre of AcMNPV-Syn-mC1-1.1 was 7.45 × 10^5^ TCID_50_/mL, which was significantly lower than that of the parental virus (*p* < 0.01). 

### 3.3. Restriction Enzyme Analysis and Genome Sequencing of AcMNPV-Syn-mC1-1.1

To verify the accuracy of the final sequence of the AcMNPV-Syn-mC1-1.1 genome, a restriction enzyme digestion assay was performed. The predicted physical maps for BglII, XhoI, and SacII digestion of AcMNPV-Syn-mC1-1.1 and AcMNPV-WIV-Syn1 are shown in Figure 4A. The electrophoresis profiles of the digested AcMNPV-Syn-mC1-1.1 genomic DNA showed the expected pattern (Figure 4C), as predicted by computer simulation (Figure 4B). In addition, genome sequencing of AcMNPV-Syn-mC1-1.1 was carried out using high-throughput sequencing, and the results showed that apart from the designed modification, there were no additional changes in the synthetic genome.

### 3.4. One-Step Growth Curve Analysis and Electron Microscopy of AcMNPV-Syn-mC1-1.1 in Comparison with the Parental Virus

To further compare the replication dynamics of AcMNPV-Syn-mC1-1.1 and that of the parental virus AcMNPV-WIV-Syn1, one-step growth curve assays were performed on Sf9 cells using an MOI of 1 (Figure 5A). The propagation of AcMNPV-Syn-mC1-1.1 appeared to reach the stationary phase after 48 h p.i., as viral titres were 1.08 × 10^5^, 1.88 × 10^5^, and 6.81 × 10^5^ TCID_50_/mL at 48 h, 72 h, 96 h p.i., respectively. The parental virus AcMNPV-WIV-Syn1 exhibited optimized productivity, with 1.15 × 10^5^, 2.10 × 10^7^, and 6.84 × 10^7^ TCID_50_/mL at 48 h, 72 h, and 96 h p.i., respectively. Statistical analysis showed that the replication dynamics of the two viruses were significantly different (*p* < 0.001). 

The morphology of AcMNPV-Syn-mC1-1.1-infected cells was compared with that of AcMNPV-WIV-Syn1 cells by transmission electron microscopy (Figure 5B). At 24 h p.i., most cells infected with AcMNPV-WIV-Syn1 showed typical viral infections, including nuclear enlargement and assembled nucleocapsids in the ring zone region and at the virogenic stroma in the nucleus, while very few nuclear enlargements and assembled nucleocapsids were observed in the samples of AcMNPV-Syn-mC1-1.1. At 48 h and 72h p.i., AcMNPV-WIV-Syn1-infected cells showed typical phenomena of late-stage infection, where occlusion bodies with multi-nucleocapsid ODVs accumulated at the ring zone of the nucleus, while AcMNPV-Syn-mC1-1.1 showed enlarged nuclei and nucleocapsids assembled around the ring zone. Notably, the *polh* gene was deleted in AcMNPV-Syn-mC1-1.1, and, as expected, no occlusion bodies were observed. These results suggest that compared to the parental virus, the viral replication process of AcMNPV-Syn-mC1-1.1, was delayed.

## 4. Discussion

Deleting non-essential genes from the baculovirus genome can improve the insertion capacity of BEVS. This study focused on reducing the C1 region of AcMNPV and successfully rescued the infectious AcMNPV-Syn-mC1-1.1, which deleted approximately 17 kb. This is an important first step towards generating a minimised AcMNPV genome. The modified C1 fragment was built on the modified B1, B2, and B3 fragments using the design-build-test approach. Our initial design (version 1.0) contained only essential genes, based on previous transcriptomics [17] and individual gene knockout assays [14]. However, the resulting AcMNPV-Syn-mC-1.0, could not produce infectious BV. This suggests that some of the non-essential genes may have synergistic effects in that they cannot be deleted together. Redesigning yielded AcMNPV-Syn-mC1-1.1 that was capable of producing infectious BVs. 

Compared to AcMNPV-Syn-mC1-1.0, AcMNPV-Syn-mC1-1.1 contained eight extra genes (*ac12, ac13, egt, odv-e26, ac17, ac26, iap-1, lef-6*). Among these, five genes or their homologues have been found to affect BV production when deleted. For example, the deletion of *Bm5*, a homologue of *ac13,* caused a lower titre of BV [20]. Similarly, viruses with the deletion of *Bm17* which is a homologue of *ac26*, showed a slow spread in cell culture [21]. The knockout of both *odv-e26* and *ac17* resulted in reduced BV levels [22]. In addition, late and very late transcription was delayed in cells infected with the *lef-6*-null Bac-mid [23]. Our results also suggest that these genes are essential to efficient BV production in the progeny. Three genes *ac12*, *egt*, and *iap-1* were retained in AcMNPV-Syn-mC1-1.1 owing to the inconvenience of their deletion during synthesis, although they had no effect on BV production when deleted individually [14,24,25].

AcMNPV-Syn-mC1-1.1 removed a total of approximately 17 kb, including 20 genes as well as two homologous repeat regions *hr1* and *hr2* in the C1 region (Table 3). The deleted genes included *ac154, bro, ctl, ac4, ac5, orf603, polh, arif-1, ac22, F, ac29, ac30, sod, fgf, lef-12, gta, ac43, ac44, ac45,* and *odv-e66.* All these genes have been found to be non-essential for BV production when deleted individually [14]. The deleted sequences covered approximately 21.5% of the C1 region (~46.4 kb), indicating the flexibility of the AcMNPV genome for minimisation and manipulation. 

The current AcMNPV-Syn-mC1-1.1, may not be the most minimalised version of C1 in terms of BV production. The deletion of additional genes in C1 while maintaining the acquisition of an infectious virus may be possible. However, as the BV titre of AcMNPV-Syn-mC1-1.1 is already significantly lower than that of the parental virus, further deletion may not be necessary because our goal was to synthesise an entirely modified genome containing modified C2 and C3. The low growth rate of AcMNPV-mC1-1.1 may be due to the modification of mB2-1.1, because the titre of AcMNPV-mB2-1.1 was significantly lower than that of the parental virus (Table 2). We will continue to improve C1 modification and modify C2 and C3 fragments to construct a modified AcMNPV, which has a much smaller genome size but retains efficient infectivity.

## Figures and Tables

**Figure 1 viruses-14-02780-f001:**
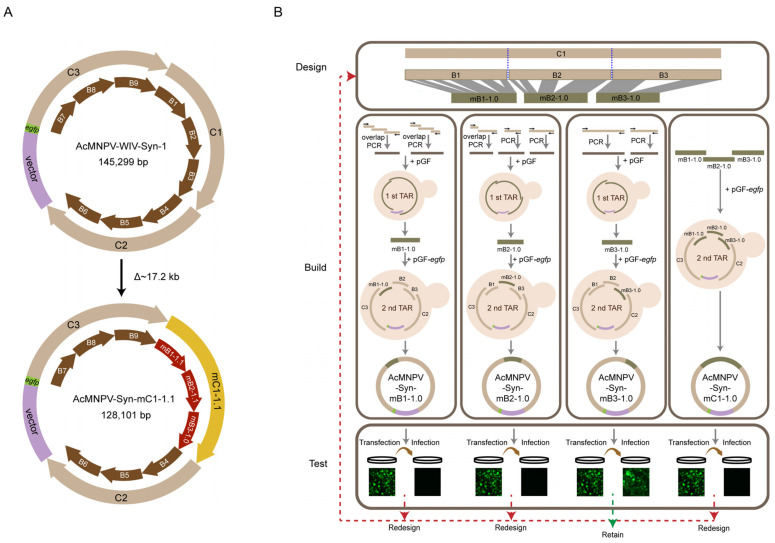
Flowchart of constructing AcMNPV genome containing modified C1 fragment. (**A**) Circular maps of the parental genome of AcMNPV-WIV-Syn1 and the modified AcMNPV-Syn-mC1-1.1. (**B**) The design-build-test cycling method to modify C1 fragment. The design of the modified C1 was based on the modification of B1, B2, and B3 fragments. Non-essential genes of each fragment were removed to generate modified fragments of mB1-1.0, mB2-1.0, and mB3-1.0. mB-1.0, mB2-1.0, and mB3-1.0 by overlapping PCR and used to generate modified genomes of AcMNPV-Syn-mB1-1.0, AcMNPV-Syn-mB2-1.0 and AcMNPV-Syn-mB3-1.0, respectively, using two steps of TAR in yeast. Meanwhile, AcMNPV-Syn-mC1-1.0 was generated using mB1-1.0, mB2-1.0, and mB3-1.0. To test, transfection and infection were performed; and the modified genome, which could not generate efficient progeny viruses, was redesigned and subjected to the design-build-test cycle again.

**Figure 2 viruses-14-02780-f002:**
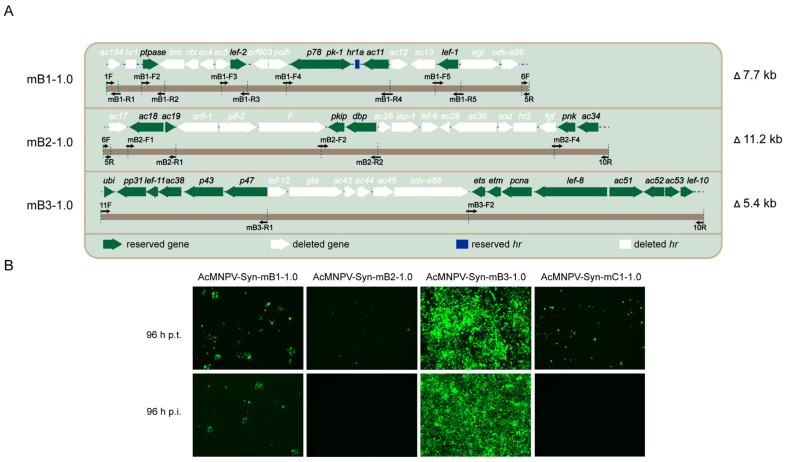
Design and characterization of version 1.0 of the modified genomes. (**A**) Design of mB1-1.0, mB2-1.0, and mB3-1.0. The modified fragments are shown on the backbones of the original B1, B2, and B3. The deleted genes and *hr*s are coloured white, the reserved genes green, and the reserved *hr1a* blue. The primers used for overlap extension PCR are indicated. (**B**) Transfection and infection results of AcMNPV-Syn-mB1-1.0, AcMNPV-Syn-mB2-1.0, AcMNPV-Syn-mB3-1.0, and AcMNPV-Syn-mC1-1.0. The images were obtained at 96 h post-transfection (p.t.) and 96 h post-infection (p.i.). The green dots represent infected Sf9 cells.

**Figure 3 viruses-14-02780-f003:**
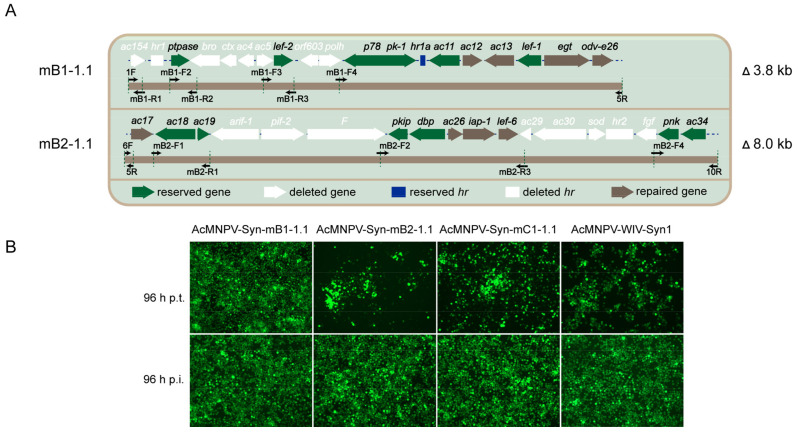
Redesign and characterization of AcMNPV-Syn-mB1-1.1, AcMNPV-Syn-mB2-1.1, and AcMNPV-Syn-mC1-1.1. (**A**) Design of mB1-1.1 and mB2-1.1. The modified fragments are shown on the backbones of the original B1 and B2. The colouring is similar to that in Figure 2A, with the eight repaired ORFs showed in brown. (**B**) Transfection and infection results of AcMNPV-Syn-mB1-1.1, AcMNPV-Syn-mB2-1.1, AcMNPV-Syn-mC1-1.1, and AcMNPV-WIV-Syn-1. The images were obtained at 96 h p.t. and 96 h p.i. The green dots represent infected Sf9 cells.

**Figure 4 viruses-14-02780-f004:**
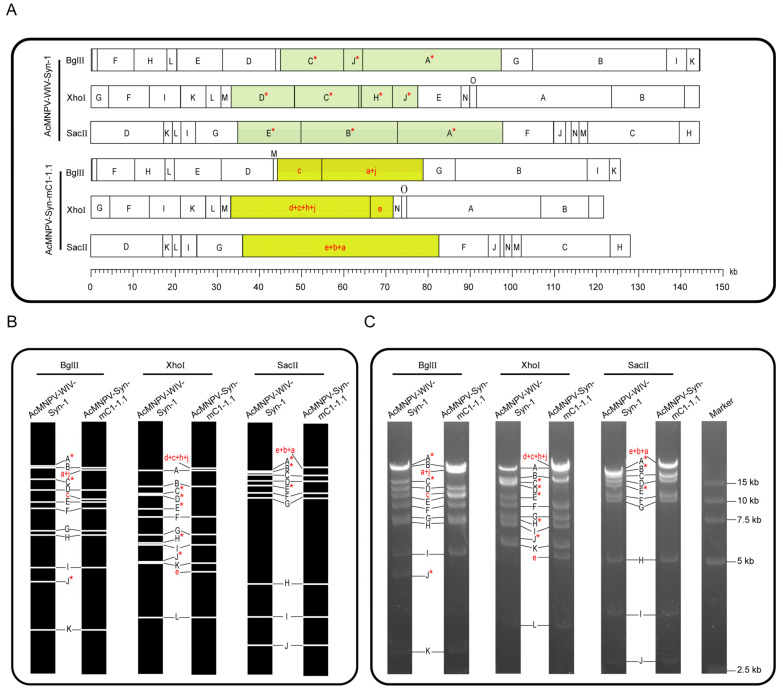
Restriction enzyme analyses of the genomes of AcMNPV-Syn-mC1-1.1 and AcMNPV –WIV-Syn1. (**A**) Physical maps of the AcMNPV-Syn-mC1-1.1 and AcMNPV-WIV-Syn1 genomes digested with BglII, XhoI or SacII. The fragments were named alphabetically according to size. The fragments corresponding to the C1 region in AcMNPV-WIV-Syn1 and the modified C1 region in AcMNPV-Syn-mC1-1.1 are highlighted. (**B**) Computer-simulated restriction enzyme profiles of AcMNPV-WIV-Syn1 and AcMNPV-Syn-mC1-1.1. (**C**) Electrophoresis results of the restriction enzyme-digested DNA samples of AcMNPV-WIV-Syn1 and AcMNPV-Syn-mC1-1.1 in 1% agarose gel. Fragments with size change are indicated in red letters in AcMNPV-Syn-mC1-1.1 and the original fragments in AcMNNPV-WIV-Syn1 are marked with a red star.

**Figure 5 viruses-14-02780-f005:**
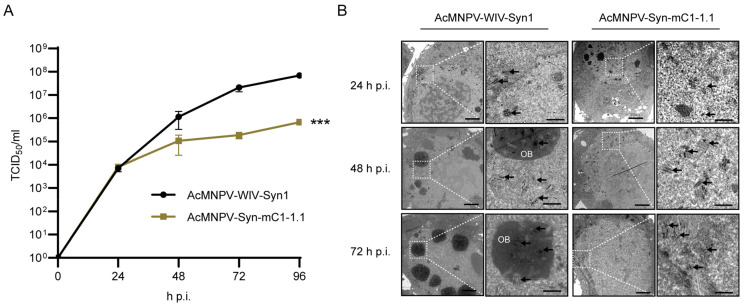
Characterization of AcMNPV-Syn-mC1-1.1. (**A**) One-step growth curves of AcMNPV-WIV-Syn1 and AcMNPV-Syn-mC1-1.1. Sf9 cells were infected with AcMNPV-WIV-Syn1 or AcMNPV-Syn-mC1-1.1 at an MOI of 1, the supernatants were collected at the indicated time points and viral titres were titrated by EPDA. The experiment was carried out in triplicate, and the titre was transformed logarithmically after averaging. Statistical significance was determined by 2way ANOVA. ***: *p* < 0.001. (**B**) Results of electron microscopy. Sf9 cells were infected with AcMNPV-WIV-Syn1 or AcMNPV-Syn-mC1-1.1 at an MOI of 5, and cells were collected at 24, 48, and 72 h p.i. for electron microscopy. The representative nucleocapsids are marked with arrows. OB: occlusion body. Scale bars are 2 μm and 1 μm.

**Table 1 viruses-14-02780-t001:** Primers used in this study.

Name	Sequence (5′-3′) *	Applied for
1F	CGACAGCTGATAGAAGAAAATACTCGTCTC	mB1-1.0 and mB1-1.1
mB1-R1	GCTTTACGAGTAGAATTCGGGGTCAACATC GATAGTGTCATACG	mB1-1.0 and mB1-1.1
mB1-F2	GAATTCTACTCGTAAAGCGAGTTGAAGGATCA	mB1-1.0 and mB1-1.1
mB1-R2	GGCCAAAAGACGTACGTGGAAAAC	mB1-1.0 and mB1-1.1
mB1-F3	GTTTTCCACGTACGTCTTTTGGCCCCATCCA ATCGACCGTTAGTCG	mB1-1.0 and mB1-1.1
mB1-R3	GGTGCGTCTGGTGCAAACTCACAAATGTTC TTGTTGCGTTTGGTT	mB1-1.0 and mB1-1.1
mB1-F4	AACCAAACGCAACAAGAACATTTGTGAGTT TGCACCAGACGCACC	mB1-1.0 and mB1-1.1
mB1-R4	GGGACGGCGATCAGCACGCGTGCCTGTAGA TCAGT	mB1-1.0
mB1-F5	GCTGATCGCCGTCCCTGTAGCATTTTGCGTT CGTGTCG	mB1-1.0
mB1-R5	CTATCAAAGCCATCTCCGCATTTACAGCAGT AAGCGTAGA	mB1-1.0
5R	CCGTCAAACGTTACATGCTTTTCG	mB1-1.0 and mB2-1.0
6F	AGATGGCTTTGATAGCGCTTATATTCAG	mB1-1.0 and mB2-1.0
mB2-F1	CGAAAAGCATGTAACGTTTGACGGTTATCG CGCAGGCGATCTTC	mB2-1.0
mB2-R1	CCGCCCATGTGTTTTAGCGAAGAGT	mB2-1.0
mB2-F2	AAAACACATGGGCGGAGCGATGATATGCCA AAACCATTGTACC	mB2-1.0 and mB2-1.1
mB2-R2	AAACTTTCTCAACTACGGGCTTGAAAGGGG CGCATTTGGAATGA	mB2-1.0
mB2-R3	GACAAAACTTTCTCAACTACGGAATAGACAAT	mB2-1.1
mB2-F4	CCGTAGTTGAGAAAGTTTTGTCCCACAGCAAA CTGGCGCTTTTATA	mB2-1.0 and mB2-1.1
10R	CACACCACGAATTATTTCCCTTCAAC	mB2-1.0
11F	AGTGGCCCGGTGTTATATTAAGTCG	mB3-1.0
mB3-R1	TGTTGCGTGCAATAGCCCTGC	mB3-1.0
mB3-F2	GCAGGGCTATTGCACGCAACACAAAAGACT GACTGTTAACA CAAAAGACTGA	mB3-1.0
16R	AAACGCTCATGTTGTGTTCGCC	mB3-1.0

*: The designed overlapping sequence is underlined, while the others are the original genome sequences.

**Table 2 viruses-14-02780-t002:** Summary of the synthesised genomes and the rescued viruses.

Name	Reduced Size (kb)	No. of Deleted Gene	Modified Position	BV Titre ^a^	
				(TCID50/mL)	*p* Value
AcMNPV-Syn1	0	0	none	6.22 × 10^7^	-
Syn-mB1-1.0	7.7	12	B1	-	-
Syn-mB1-1.1	3.8	8	B1	6.39 × 10^7^	0.9998 (ns)
Syn-mB2-1.0	11.2	12	B2	-	-
Syn-mB2-1.1	8	8	B2	4.58 × 10^6^	0.0093 (**)
Syn-mB3-1.0	5.4	6	B3	5.65 × 10^7^	0.9839 (ns)
Syn-mC1-1.0	24.2	30	B1, B2, and B3	-	-
Syn-mC1-1.1	17.2	22	B1, B2, and B3	7.45 × 10^5^	0.0062 (**)

^a^ The Sf9 cells were infected with the indicated viruses at an MOI of 1. The supernatants of the infected cells were collected at 96 h p.i. and the BV titre was measured using EPDA. Each experiment was performed in triplicates. -: not applied. Statistical analysis was performed using single factor analysis of variance. ns: not significant; **: *p* < 0.01.

**Table 3 viruses-14-02780-t003:** Gene deletion in AcMNPV-Syn-mC1-1.1.

Number	Deleted Gene	Protein	Position	Function	Impact on BV Production
1	*ac154*		B1	Unverified	Non-essential [14]
2	*hr1*		B1	Enhancer [18]	Non-essential [26]
3	*bro*	Baculovirus repeated orf	B1	Unverified	Non-essential [27]
4	*ctl*	Conotoxin-like protein	B1	Inhibitor of melanisation [28]	Non-essential [29]
5	*ac4*		B1	Unverified	Non-essential [30]
6	*ac5*		B1	Unverified	Non-essential [31]
7	*orf603*	Orf 603	B1	Unverified	Non-essential [32]
8	*polh*	Polyhedrin	B1	Occlusion body matrix protein [33]	Non-essential [34]
9	*arif1*	Arif-1	B2	Induction of actin rearrangement [35]	Non-essential [36]
10	*pif2*	PIF-2	B2	Oral infection [37]	Non-essential [37]
11	*F*	Fusion protein	B2	Associated with cell binding [38]	Non-essential [39]
12	*ac29*		B2	Unverified	Non-essential [14]
13	*ac30*		B2	Unverified	Non-essential [40]
14	*sod*	Super oxide dismutase	B2	Super oxide dismutase [41]	Non-essential [41]
15	*hr2*		B2	Enhancer [18]	Non-essential [26]
16	*fgf*	Fibroblast growth factor	B2	Stimulation of insect cell motility [42]	Non-essential [42]
17	*lef-12*	Lef-12	B3	Transcription [43]	Non-essential [44]
18	*gta*	Global transactivator-like protein	B3	Unverified	Non-essential [14]
19	*ac43*		B3	Affect polyhedron [45]	Non-essential [45]
20	*ac44*		B3	Unverified	Non-essential [14]
21	*ac45*		B3	Unverified	Non-essential [14]
22	*odv-e66*	ODV-E66	B3	Oral infection [46]	Non-essential [46]

## Data Availability

The data is available within the manuscript.

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
