# Peer review of "Successful Rescue of Synthetic AcMNPV with a ~17 kb Deletion in the C1 Region of the Genome"

_viruses, 2022, doi:10.3390/v14122780_

Round 1

Reviewer 1 Report

As widely used protein expression vectors, baculoviruses deserve great attention on their gene functions. The authors­ paid efforts to elucidate which genes in the genome of baculovirus are essential and which are not. It is interesting work, but there are still some issues as followings listed should be addressed.

1. In 3.2, what is the principle for selection of genes to be repaired while redesign the three viruses? Why did you choose ac12, ac13, egt, odv-e26 to mB1-1.0 and ac17, ac26, iap-1, lef-6 to mB2-1.0 instead of other genes?

2. As is shown in Table.2, The titers of AcMNPV-Syn-mB1-1.1 and AcMNPV-Syn-mB3-1.0 were not significantly different from those of the parental virus, so it seems that the modification on B2 accounts for the decline of titers. Why not try a further redesign of AcMNPV-Syn-mB2-1.1?

3. Although the AcMNPV-Syn-mC1-1.1 was infectious, its titer was significantly lower than that of the parental virus. Therefore, AcMNPV-Syn-mC1-1.1 is still not a successful improvement and further repair is needed.

4. Besides the cell lines, it is better to test the viral infectivity in insect individuals.

5. The priority goal of the improvement of baculovirus is to increase the expression of target genes. The study did not conduct any experiment to investigate whether the modifications made on viral genome influence the expression efficiency of the target gene at polh loci.

Author Response

We thank the reviewer for the positive opinion on the paper. We have responded to the comments and suggestions. Please see the attachment.

Reviewer 2 Report

This is a very important work, which makes it possible to further opimize the BEV expression system.

There are still some questions that I would like to propose. The authors should describe the construction method in yeast by TAR in short. Moreover, as we know baculovirus could entry into mammalian cells in some conditions, so is it feasible if perform this study in mammalian cells?

Author Response

(The authors gave the same response as above.)

Reviewer 3 Report

Baculoviruses have a circular double-stranded DNA (dsDNA) genome with 90 to 180 genes, some of which are nonessential genes. As previous research proved that disruption of nonessential genes will improve production of baculoviral expression system, the deletion of nonessential genes from the baculovirus genome to make it a more optimized vector. In this article the authors used synthetic biology techniques to minimize AcMNPV genome, and finally get a synthetic AcMNPV with a 17 kb deletion. The methodology used for this manuscript is properly and clearly described. Data have been correctly interpreted and conclusions are sound. The manuscript is well prepared, only a few issues are need to be improved before considering the manuscript for publication:

1.     It may be beneficial for the authors to include a table of deleted genes in AcMNPV-Syn-mC1-1.1 and their putative functions in the manuscript or supplementary materials

2.     Why the term “egfp” in “pGF-egfp” is underlined in line 101 and line 104?

3.     What is the green bar indicated in the circular map in figure1A?

4.     Species names should be italic in reference

Author Response

(The authors gave the same response as above.)
